# Cryopreservation of Holm Oak Embryogenic Cultures for Long-Term Conservation and Assessment of Polyploid Stability

**DOI:** 10.3390/plants11091266

**Published:** 2022-05-08

**Authors:** Maria Teresa Martínez, Sonia Suárez, Paloma Moncaleán, Elena Corredoira

**Affiliations:** 1Misión Biológica de Galicia (MBG-CSIC), Sede Santiago de Compostela, Avda de Vigo s/n, 15705 Santiago de Compostela, Spain; temar@mbg.csic.es; 2Neiker-BRTA, Centro de Arkaute, Campus Agroalimentario de Arkaute, 01213 Arkaute, Spain; ssuarez@neiker.eus (S.S.); pmocalean@neiker.eus (P.M.)

**Keywords:** cryobiotechnology, flow cytometry, genetic stability, oak decline, plant vitrification solutions, *Quercus ilex*, somatic embryogenesis, vitrification

## Abstract

Holm oak populations are severely affected by oak decline syndrome, and reliable methods of conserving the plant material are required. A vitrification-based cryopreservation method was used for the first time for the long-term conservation of holm oak embryogenic cultures. Successful cryopreservation was achieved after determining the best developmental stage of the somatic embryos used and the optimal incubation period in plant vitrification solution 2 (PVS2). Embryos were recovered from individual nodular embryogenic structures (NES) derived from four embryogenic lines after preculture on a medium containing 0.3 M sucrose, incubation in PVS2 vitrification solution for 15 min at 25 °C and direct immersion in liquid nitrogen (LN). Embryo recovery rates of 16.7–63.3% were obtained after cryostorage for four years in LN. In addition to the embryo developmental stage and the PVS2 treatment time, the genotype can also significantly affect embryo recovery after LN storage. There were no significant differences in plant regeneration or polyploid stability between somatic embryos and plants derived from control embryos (not cryopreserved) and cryopreserved embryos. The findings indicate that embryo proliferation, plant conversion and polyploid stability are maintained in material recovered from the vitrification solution and subsequently cryopreserved.

## 1. Introduction

In the Mediterranean basin, holm oak woodlands and dehesas (grazed sclerophyllous forest systems exclusive to the Iberian Peninsula) are of great economic importance and play a crucial role in maintaining ecosystem services and biodiversity. Mediterranean holm oak forests have been altered by human activity over centuries and have been overworked, modified or converted to agricultural and/or urban land [1]. In recent years, climate change has also contributed to altering the genetic resources in these ecosystems [2,3]. According to the Intergovernmental Panel on Climate Change 2021 report, an increase in aridity due to more frequent episodes of droughts and high temperatures will occur [4]. This phenomenon will lead to degradation, desertification and overexploitation of the forests, directly affecting the growth rate of trees [5], altering the distribution area of pathogens and pests [6,7] and increasing fire risk [8]. These effects will seriously threaten the future of holm oak woodlands and the dehesas, and immediate action is required to ensure the conservation of these ecosystems and the diversity of holm oaks [9].

The ex situ conservation of *Quercus ilex* L. germplasm, as with other oak species, is complex due to the recalcitrant nature of their seeds [10]. Currently, the seeds can only be stored without losing their viability during the spring after harvesting or a few months later [11]. Biotechnological tools such as in vitro culture and cryobiotechnological techniques provide long-term storage strategies to conserve the germplasm of species with recalcitrant seeds [9,10,12]. These technologies have also been used successfully to conserve plant material derived from biotechnological processes such as genetic transformation and somatic embryogenesis (SE). Moreover, creating banks of biotechnologically-derived germplasm complements other ex situ conservation methods such as storage of field collections, which are potentially more susceptible to loss of diversity due to natural disasters and the spread of disease.

Cryopreservation is currently considered a safe, cost-efficient method for the long-term storage of plant germplasm [9,13,14]. The method involves the storage of live tissues at ultra-low temperatures in liquid nitrogen (LN) (−196 °C) and/or its vapor phase (−150 °C) [9,15]. Among the different ex vitro or in vitro plant tissues that can be used to cryopreserve hardwood species, somatic embryos are considered the best material for subsequent regeneration [15]. The combination of SE and cryopreservation is especially useful in forest breeding programs, as the ability to select superior genotypes can be greatly refined, which is the basis of multivarietal forestry [16]. In addition, cryopreservation also prevents the loss of embryogenic capacity and the appearance of putative somaclonal variations, reduces costs associated with regular subculturing and minimizes the risk of loss of plant material due to contamination or technical or human error [15,17,18]. The main prerequisite for successful cryopreservation is that the water content of the cells must be reduced in order to prevent the formation of ice crystals during cooling; the embryos must also be sufficiently dehydrated and/or treated with cryoprotective solutions to ensure correct development after storage in LN [19]. Procedures available for the cryopreservation of somatic embryos of forest species include slow freezing, desiccation, pre-growth drying, encapsulation-dehydration, vitrification, encapsulation-vitrification and droplet vitrification [13,15]. Vitrification-based procedures are commonly used to reduce water content and cryoprotect cells because they are operationally less complex than classical slow freezing (as they do not require the use of controlled freezers), and they are of potentially broad application, requiring only minor modifications for different cell types [20]. Briefly, vitrification involves the direct transition of water from a liquid to a vitreous or amorphous stage by the treatment of plant material with highly concentrated and viscous cryoprotectant solutions [21]; the vitrification solutions comprise concentrated mixtures of penetrating and non-penetrating cryoprotectants [22]. The frequently used plant vitrification solution (PVS2) was developed by Sakai et al. [23] to induce cell vitrification of navel orange (*Citrus sinensis* Osb.) nucellar cells [13,24]. 

Vitrification has been used successfully for the cryopreservation of zygotic and somatic embryos of different species [15,25,26], including species of *Fagaceae* such as European chestnut [27] and some species of the genus *Quercus* [10,28,29,30]. However, to date, little research has focused on the cryopreservation of holm oak, as reflected in the very small number of publications on the subject: a protocol for the cryopreservation of somatic embryos that did not allow storage beyond 24 h [31], and a cryopreservation protocol for zygotic embryo axes, in which germination/regeneration after storage was not successful [32].

The current study focused on the development of an effective long-term vitrification-based storage method for holm oak embryogenic cultures. In addition, polyploid stability in the somatic embryos and in plants regenerated after cryopreservation was analyzed and addressed. With this main objective, we hope to increase the diversity of the genetic resources of this species and guarantee their maintenance for future generations. 

## 2. Results

### 2.1. Effect of Explant Type

The aim of the first experiment was to determine the best developmental stage of somatic embryos for cryopreservation in LN. For this purpose, three types of explants isolated from E2 and Q8 embryogenic lines, i.e., nodular embryogenic structures (NES), globular-heart stage embryos and early cotyledonary stage embryos (Figure 1a–c), were evaluated. 

In both lines, the survival rate was significantly influenced by the developmental stage of somatic embryos, cryostorage in LN and the interaction of both factors (*p* ≤ 0.001) (Table 1). As there was a positive interaction between factors, each factor was analyzed separately. Survival rates in the material treated with PVS2 but not cryopreserved were not significantly different, being 100% in all explant types. By contrast, cryostorage in LN negatively affected survival; thus, only the E2 NES survived cryostorage in LN (80%), while in line Q8, both NES and globular embryos survived, with the survival rate of NES being significantly higher (88.3%) (Table 1). Similarly, in both lines, the embryo recovery rate was also affected by the type of explant (*p* ≤ 0.001), cryostorage in LN (*p* ≤ 0.001) and the interaction between these factors (*p* ≤ 0.01) (Table 1). In both lines, the best results were obtained with NES immersed in LN. Although line Q8 new embryos developed when NES or globular embryos were used as the initial explants, recovery of NES was significantly higher (56.7%). In line E2, embryo recovery was only achieved when NES were used as the initial explants (46.7%). 

Treatment with PVS2 seemed to negatively affect embryo recovery in the globular and cotyledonary stage embryos. In embryos at these developmental stages treated with PVS2 and not cryopreserved, the development of new embryos was also significantly lower, especially at the early-cotyledonary stage. Finally, in both embryogenic lines and in all explant types evaluated, in both controls, i.e., explants immersed in LN but which had not undergone preculture on medium containing 0.3 M sucrose and treatment with PVS2 solution (overall control) or which were only cultured on medium containing 0.3 M sucrose (Suc 0.3 M control), the survival rate was zero (data not shown).

### 2.2. Effect of PVS2 and LN Cryostorage Period

The aim of the second experiment was to optimize the length of treatment with PVS2 solution (15 or 30 min) and to determine whether the solution was equally effective for both embryogenic lines (Q8 and E2). At the same time, we also examined the effect of cryostorage in LN for 1, 3 or 12 months to determine whether holm oak embryogenic cell line cultures can be preserved for long periods (Table 2a,b). 

Explant survival was observed in both lines, with two PVS2 incubation periods and in the three LN cryostorage times evaluated. In both embryogenic lines, genotype and PVS2 treatment significantly affected (*p* ≤ 0.05) the survival of controls, but not their interaction. In both lines, survival rates were higher than 90% for precultured controls treated with PVS2 for 15 min, but not cryopreserved. This finding indicates that PVS2 did not have a toxic effect on the somatic embryos for this short treatment period (Table 2a). However, in line E2, treatment with PVS2 for 30 min reduced the survival percentage (60%), indicating possible toxic effects. In this line, in cryopreserved explants, the best results were also obtained after treatment with PVS2 for 15 min in all storage times evaluated, although the survival rate was lower than in line Q8. In contrast, line Q8 survival was higher for incubation in PVS2 for 15 min after cryostorage in LN for one month; however, after cryostorage in LN for 6 and 12 months, the best results were obtained with explants treated with PVS2 for 30 min, although the differences were not significant. 

Embryo recovery in PVS2-treated explants without LN followed a similar trend as survival (Table 2b). In line Q8, embryo formation was elevated in the two PVS2 treatments tested, whereas in line E2, incubation in PVS2 for 30 min reduced embryo formation (Table 2b). In cryopreserved material, embryo recovery was observed in both embryogenic lines, and again better results were obtained with line Q8. Thus, recovery rates in line Q8 were higher than 50% for both periods of PVS2 treatment and all three LN cryostorage times, although PVS2 treatment for 15 min and one-month cryostorage in LN yielded the best results (83.3%). In line E2, embryo recovery rates were higher in all cases for the 15 min PVS2 treatment, ranging from 53.3 to 60% (Table 2). In cryopreserved NES of both embryogenic lines, explants appeared blackish in color during the first weeks, and new somatic embryos arose at a variable rate, appearing as early as the third week after cryopreservation (Figure 1d). 

In view of these results, treatment with PVS2 for 15 min was considered best for cryopreserving holm oak embryogenic cultures. 

### 2.3. Effect of Genotype

When the previously defined procedure was used to cryopreserve NES from four embryogenic lines, survival after one month was not significantly affected by the genotype but was influenced by cryostorage in LN (*p* ≤ 0.001) and by the interaction of both factors (*p* ≤ 0.05) (Table 3). 

The survival rate of the material treated with PVS2 but not cryopreserved reached 86–90% in all embryogenic lines (Table 3). However, the survival rates were significantly lower in cryopreserved explants of all lines, except line Q8, in which the survival rate was 80%. Similarly, the embryo recovery rate was also significantly affected by cryostorage in LN (*p* ≤ 0.001), but not by genotype, and in this case, there was no significant interaction between the two factors (Table 3). The embryo recovery rate in explants treated with PVS2 but not cryopreserved was 80% for all lines evaluated. By contrast, in the four lines of explants cryostored in LN, the embryo recovery rates were significantly lower (*p* ≤ 0.001), with values ranging between 36.7 and 60%.

### 2.4. Effect of Long-Term Cryopreservation and Genotype

The previously described protocol was used to cryopreserve the explants of four lines in LN for four years (Figure 1e). After this period, although the generation of new somatic embryos occurred in the four lines, survival and embryo recovery were significantly (*p* ≤ 0.001) affected by the genotype (Figure 2). 

Survival rates were significantly higher in lines E00 and Q8, reaching values of 66.7% and 60%, respectively. A similar trend was observed for embryo recovery rates, and the same lines yielded the highest percentages, with values higher than 50%. These values contrast with those obtained in lines Q10–16, in which the embryo recovery rate was only 16.7% (Figure 2).

### 2.5. Plant Regeneration

Embryogenic cultures of Q8 and E2 lines pretreated with PVS2 solution for 15 min at 25 °C were recovered from LN after one year. After thawing the embryos and culturing them for 8 weeks on proliferation medium, cotyledonary embryos were isolated for maturation by cold storage at 4 °C. The embryos of both lines converted into plantlets with developed roots and shoots at rates of 54.2% for line Q8 and 45.8% for line E2 (Table 4; Figure 3). The plants developed had adequate morphology and were of good quality in terms of leaf number and shoot length (Table 4). No phenotypic abnormalities were detected when plantlets derived from cryostored embryos when compared with those derived from embryos that were not cryopreserved. 

### 2.6. Analysis of Ploidy Stability by Flow Cytometry

Flow cytometry analysis was used to assess the genetic stability (in terms of polyploidy) of cryopreserved embryos and derived plantlets. All of the samples analyzed yielded two peaks in stable positions in the PI-fluorescence histograms corresponding to nuclei (2C) in phase Go/G1 of *Q. ilex* and *P. sativum*. No additional peaks were observed, thus ruling out the existence of polyploidy in the cryopreserved material.

The DNA index (DI) and the corresponding value of nDNA content are shown in Table 5. The DI obtained for embryos cryopreserved in LN was very similar to that of non-cryopreserved embryos in both embryogenic lines (0.244–0.243 for E2 and 0.216–0.219 for Q8). For plantlets, DI was also similar for the material obtained from cryopreserved (0.247 for E2 and 0.233 for Q8) and non-cryopreserved lines (0.237 for E2 and 0.226 for E8). Statistical analysis confirmed no significant differences between the material derived from cryopreserved and non-cryopreserved embryogenic lines (*p* ≤ 0.05). The mean value for the coefficient of variation (CV) was slightly higher than 5% in all the samples obtained from *Q. ilex* plant material.

These results extend to the nDNA content, determined by multiplying the DI by the genome size of the internal standard plant material. However, it is worth noting that the genome was slightly larger in line E2 than in line Q8 (range between 2.12–2.24 and 1.97–2.12 pg/2C, respectively), although the difference between embryogenic lines was also statistically non-significant. 

## 3. Discussion

Cryobiotechnology constitutes a valuable tool for conserving plant germplasm, especially of woody species with recalcitrant seeds, such as holm oak [10]. Interest in this species has increased in recent years because it is severely affected by oak decline syndrome, which has prompted research into developing methods for selecting and propagating tolerant genotypes and for conserving the selected lines. Clonal forestry breeding based on cryopreservation and somatic embryogenesis can provide many benefits. Our research team has published several protocols for inducing somatic embryogenesis for cloning adult holm oak trees [33,34]. Cryopreservation of these embryogenic lines appears to offer the best prospects for the long-term conservation and management of valuable holm oak genotypes. Although considerable progress has been made in the last two decades in the cryopreservation of somatic embryos of many woody species, including several species of the *Fagaceae* family [10,15,30,35,36], the cryopreservation of holm oak embryogenic cultures has received scant attention. Barra-Jiménez et al. [31] applied a vitrification-based method to cryopreserve two of the four lines used in the present study; however, embryogenic recovery was only successful with explants stored for 24 h in LN, while embryos stored for one month produced callus, and embryo recovery was not observed. In the present study, an efficient protocol for the long-term cryopreservation of oak embryogenic lines has been defined for the first time by reducing the time of treatment with PVS2 solution from 30 to 15 min and using isolated NES instead of globular or heart stage embryos as explants.

The age, size and stress tolerance of the explant are critical factors for the success of cryopreservation [9,37]. In the case of somatic embryos, recovery of cryopreserved embryos is significantly affected by the developmental stage of somatic embryos used, and early developmental stages usually perform better [30]. In holm oak, NES withstands cryostorage in LN better than the more differentiated cotyledonary embryos, in which the cells exhibit higher levels of vacuolization and differentiation than the actively dividing cells present in the most superficial layers of the NES [30,38]. These characteristics of the nodular structures probably provide better resistance to desiccation caused by vitrification solutions. Indeed, it has been pointed out that only small dense cells, rich in cytoplasm with few vacuoles, survive cryostorage in LN in two *Picea* species [39]. In cork oak, small groups of globular or heart embryos responded better to storage in LN than more differentiated cotyledonary stage embryos [29]. In olive, organized embryogenic tissues were more susceptible to PVS2 than unorganized tissues [40]. It was similarly concluded that the successful cryostorage of somatic embryos of avocado was significantly determined by the degree of differentiation of the explants used [41]. It is also well known that explants destined for cryopreservation should show a good proliferation capacity because this is an important attribute that can favor the resumption of growth after thawing [42]. This is consistent with findings in holm oak embryogenic cultures, in which NES display the highest multiplication capacity among different types of explants [33,43]. The thawed explants were blackish in appearance during the first weeks, as also observed in other forest species such as *Castanea dentate* [44], *Liriodendron tulipifera* [45] and *Castanea sativa* [27], and is attributed to a state of cryoshock; however, signs of survival and of the formation of new embryos were observed after culture of the explants for 3–4 weeks.

Vitrification has been the most widely used method for cryopreserving somatic embryos of oak species, including pedunculate oak [28], cork oak [29] and white oak [30]. Similarly, vitrification proved more effective than rapid desiccation for the cryopreservation of European chestnut somatic embryos [27]. Moreover, vitrification has also been successfully used in the cryopreservation of transgenic somatic embryos of these species, obtained after overexpression of pathogenesis-related proteins by *Agrobacterium tumefaciens* [46,47,48,49]. Determination of the best incubation time in vitrification solution and the temperature during the incubation period is essential for vitrification-based methods [15]. It is extremely important to strike a balance between correct dehydration and the toxicity that the vitrification solutions can cause [22]. These solutions act by removing intracellular liquid from the plant tissues and inducing intracellular water to undergo a transition from a liquid phase to an amorphous phase upon rapid cooling, without forming lethal ice crystals and, hence, ensuring a high survival rate [50]. The results obtained in the present study show that PVS2 can have a toxic effect on somatic embryos of holm oak, and the optimum treatment time with PVS2 for maximum viability was found to be 15 min. The toxicity of PVS2 usually increases over time and reduces the embryo survival rate [51]. The short application period in holm oak contrasts with the periods used for the cryopreservation of somatic embryos of other oak species: 60–90 min for pedunculate oak [28,52], 60 min for cork oak [29] and 30 min for white oak [30]. However, the narrow window of time of treatment with PVS2 is consistent with the results regarding the cryopreservation of other species [53,54,55]. The aforementioned difference can probably be explained by species-specific and even cultivar-specific differences in tolerance to vitrification solutions in plant tissues [56]. Differences in the tolerance capacity can be explained by the different water content or by differences in permeability of the membrane to the cryoprotectants, and the period of incubation in the vitrification solutions must be optimized in each case [42]. Moreover, the results obtained revealed differences related to the developmental stage of somatic embryos, as embryos at globular and especially at cotyledonary stages were more sensitive to the adverse effects of treatment with PVS2. The different tolerance to vitrification solutions in relation to the developmental stage of somatic embryos has also been mentioned in other species such as *Prunus avium* [57] and *Q. suber* [29].

In holm oak, the genotype significantly affected embryo recovery after cryostorage for 4 years in LN, as high rates of growth resumption were observed in lines Q8 and E00 (63.3% and 53.3%, respectively), while the corresponding rates were lower in lines E2 and Q10–16 (30 and 16.7%, respectively). Genotype-related variations in response to cryostorage in LN have been reported for *Q. robur* [52], *Alnus glutinosa* [58] and *Persea americana* [59]. These results emphasize the importance of evaluating a wide range of genotypes to define an efficient cryopreservation procedure.

Despite the above-mentioned effects of genotype, acceptable embryo recovery rates were achieved in three embryogenic lines stored for 4 years in LN. Embryogenic recovery rates of 57–92% for pedunculate oak [28,52], 88–93% for cork oak [29] and about 54% for white oak [30] have been reported, although the somatic embryos were cryopreserved for a maximum of one year. It is important to highlight that in the present study, the embryo recovery capacity was evaluated after storage of the embryos in LN for 4 years.

Plants were regenerated from cryopreserved material with similar conversion rates as controls (not cryopreserved), and the regenerated plantlets were vigorous and showed good shoot and root growth. Likewise, Valladares et al. [29] observed similar conversion rates in cryopreserved embryos and control embryos of cork oak. By contrast, the effect of cryopreservation on plant regeneration ability of somatic embryos of olive varied significantly depending on the genotype [60]. 

Assessment of the genetic fidelity of thawed somatic embryos and regenerated plants is another important aspect of the cryopreservation process [14]. The genetic stability of cryopreserved material has been evaluated using different molecular techniques, such as phenotypic/morphological studies, biochemical and molecular data and, more recently, flow cytometry [61,62], as used in the present study. In our study, relative nuclear DNA content was not modified after cryopreservation, in agreement with similar studies carried out in several wood species, including cork oak [63], *Ginkgo biloba* [64], black alder [58] and *Thuja koraiensis* [65]. 

## 4. Materials and Methods

### 4.1. Plant Material and Culture Conditions

Four somatic embryo lines of holm oak (*Quercus ilex* L.), denominated E2, Q8, E00 and Q10–16, were used in the cryopreservation experiments. Lines E2, Q8 and E00 were induced from teguments of ovules derived from mature holm oak trees [66], while line Q10–16 was initiated from a leaf explant isolated from in vitro axillary shoot cultures established from a centennial holm oak tree [33]. Embryogenic lines are maintained by secondary embryogenesis by subculturing NES, with subculture every 6 weeks on proliferation medium consisting of SH mineral medium [67], MS vitamins [68], 30 g/L sucrose and 6 g/L PPA agar (Condalab, Torrejón de Ardoz, Spain). 

Stock embryos cultures were incubated in climate-controlled growth chambers with a photoperiod of 16/8 h light/darkness under white light fluorescent lights (Mazdafluor 7D TF 36 w/LJ) and a photonic flux density of 50–60 μmol·m^−2^·s^−1^) at 25 °C ± 1 °C in light and 20 °C ± 1 °C in darkness (standard conditions).

### 4.2. Cryopreservation Equipment

The liquid nitrogen used for cryopreservation was stored in a Thermo Scientific Locator Rack and Box unit (Locator 8, Thermo Fisher Scientific, Waltham, MA, USA) with an approximate capacity of 121 L. The cryogenic unit is lined with insulating material and has a lid to reduce evaporation of the LN. The cryogenic unit can hold 8 boxes in different chambers, and each box can store up to 25 sterile cryogenic vials (Nalgene^®^, Sigma Aldrich, St. Louis, MO, USA) of capacity 2 mL.

### 4.3. Cryopreservation Experiments

#### 4.3.1. Standard Cryopreservation Procedure

For cryopreservation experiments, somatic embryos were cultured in Petri dishes containing 25 mL of proliferation medium supplemented with 0.3 M sucrose for 3 days under standard conditions. The embryos were then placed in 2 mL cryovials (10 embryos per vial) and 1.8 mL of PVS2 vitrification solution (30% *w*/*v* glycerol, 15% *w*/*v* DMSO, 15% *w*/*v* ethyleneglycol in SH medium containing 0.4 M sucrose [23]) was added to prevent damage to cells during freezing-thawing. The embryos were incubated in PVS2 at 25 °C, and they were then placed in 0.6 mL of PVS2 solution before being plunged in liquid nitrogen (−196 °C). The embryos were thawed by immersion in a water bath at 42 °C for 2 min. The PVS2 solution was then removed, and the embryos were washed twice, for 10 min, with liquid SH medium supplemented with 1.2 M sucrose before being placed on a sterile filter paper disc previously placed in a Petri dish containing proliferation medium. Twenty-four hours later, the explants were transferred to a proliferation medium without any filter paper discs and were cultured for eight weeks under standard conditions.

#### 4.3.2. Effect of Explant Type

In a first experiment, the effect of the somatic embryo developmental stage used as the explant destined for cryopreservation was evaluated. For this purpose, three embryogenic developmental stages of lines Q8 and E2 were compared: 1–3 individual NES, groups of 2–3 globular-heart and individual early cotyledonary somatic embryos isolated from each of the stock embryogenic lines 4 weeks after the final subculture.

All explant types were precultured on medium containing 0.3 M sucrose for 3 days under standard conditions and were then placed in PVS2 for 15 min at 25 °C. In each embryogenic line, at least 6 replicates of each explant type were used: half of the samples were immersed in LN (LN+ explants) for one month, and the other half were not stored in LN (LN− explants). In addition, two controls were evaluated: an overall control, in which sucrose preculture and PVS2 treatment were not applied, and a Suc 0.3 M control, in which only preculture on sucrose-containing medium was applied. 

#### 4.3.3. Effect of PVS2 and LN Storage Time

In a second experiment, the effects of PVS2 treatment time and LN storage time were tested. Individual NES isolated from embryogenic cultures of lines Q8 and E2 were precultured on medium containing 0.3 M sucrose for 3 days under standard conditions, before being placed in PVS2 for 15 or 30 min at 25 °C and stored in LN for 1, 6 and 12 months. In each embryogenic line and for each PVS2 treatment time, 9 cryovials were immersed in LN (cryopreserved explants), and three cryovials were subsequently removed after each different LN storage time. Moreover, 3 replicates were evaluated for each incubation time in PVS2 but without LN storage (LN−). The controls described in Section 4.3.2 were also evaluated in this experiment.

#### 4.3.4. Effect of Genotype

In a third experiment, the effect of genotype of the somatic embryos of the different lines was determined. For this purpose, we evaluated the most successful cryopreservation procedure defined in the experiments described above (preculture of NES on medium containing 0.3 M sucrose for 3 days, treatment with PVS2 for 15 min at 25 °C and cryostorage in LN a month) for the cryopreservation of 4 different embryogenic lines: Q8, E2, E00 and Q10–16, maintained by secondary embryogenesis according to the conditions indicated in Section 4.1. At least 3 replicates of ten NES were used for each embryogenic line.

#### 4.3.5. Effect of Long-Term Cryopreservation and Genotype

Based on the most successful cryopreservation method, the NES of four embryogenic lines were cryopreserved for long-term storage with assessment after 4 years for all lines. At least 3 replicates of ten NES were used for each embryogenic line.

### 4.4. Plant Regeneration

Germination experiments were carried out with LN+ and LN- somatic embryos of lines E2 and Q8 following the procedure described by Martínez et al. [33]. Briefly, to promote maturation, cotyledonary-stage somatic embryos (≥ 5 mm) were placed in empty Petri dishes, where they were held for a period of two months at low temperature (4 °C) in darkness. The somatic embryos were then germinated in glass jars (500 mL) with 70 mL of GD mineral medium [69] containing 0.1 mg/L of 6-benzyladenine and 20 µM silver thiosulfate, where they were held under standard culture conditions for 8 weeks. The percentage of somatic embryos that converted into plantlets (i.e., with simultaneous shoot and root development ≥ 5 mm) was then recorded. For each line, 24 somatic embryos were cultured (6 embryos per jar). 

### 4.5. Ploidy Stability Analysis by Flow Cytometry

Flow cytometry coupled with nuclear DNA fluorescent staining was used to assess ploidy stability in embryos cryopreserved for one year and in the derived plantlets. For nuclei extraction, 20–30 mg of plant material (2–3 embryos or 2–3 young leaves) was placed in a plastic Petri dish containing 1.5 mL of Wood Plant Buffer [70]. A piece of *Pisum sativum* leaf (2C = 9.09 [71]) was added as internal reference material. The tissues were chopped with a sharp blade for 30 s to release nuclei. The nuclei suspension was then filtered to remove large debris (CellTrics^®^ 50 μm nylon filter), and 0.5 mL aliquots of the filtrate were transferred to an Eppendorf vial.

Samples were stained with 50 μL/mL of propidium iodine (PI, Fluka), a fluorochrome that penetrates the nuclei and emits fluorescence proportionally to the nuclear DNA content. RNAase (Sigma^®^, St. Louis, USA) was also added (50 μg/mL) to prevent double-stranded RNA contributing to the fluorescence. Samples were placed on ice for 2 min before analysis of the nuclei by flow cytometry.

Nuclei were analyzed in a Cytoflex flow cytometer (Beckman Coulter). Nuclei suspensions were injected at 60 mL min^−1^ and excited with a blue (480 nm) diode laser. Two signals were collected: forward scattered light (FSC) and fluorescence emission at 588 nm ± 42 nm (PI-fluorescence). Fluorescence emission was set at a primary threshold to prevent the detection of non-fluorescent particles present in the sample. Data for at least 3000 nuclei were acquired, plotted and analyzed using CytExpert 2.1 Software (Beckman Coulter). 

First, holm oak and *P. sativum* nuclei were identified and discriminated from other fluorescent particles (partial nuclei and other debris) according to their light-scattering and fluorescence signals (FSC vs. PI-fluorescence dot plot). The selected nuclei populations were analyzed in a second cytogram (FSC-area vs. FSC-high) to discriminate doublets. Finally, signals for single nuclei were displayed in PI-fluorescence histograms to analyze the peak patterns corresponding to different nuclei populations with different nDNA content. Mean fluorescence emission and the coefficient of variation (CV) were obtained for each identified peak. 

The DNA index was calculated from the relative position of the G0/G1 peaks of holm oak and the internal reference: DI = *Q. ilex* G0/G1 peak mean fluorescence/*P. sativum* G0/G1 peak mean fluorescence). Nuclear DNA content was further calculated by multiplying the DNA index by the known genome size of the internal standard: 2C nDNA content (pg) = DI × 9.09.

### 4.6. Data Collection and Statistical Analysis

In all cryopreservation experiments, the effectiveness of different treatments was evaluated by recording the explant survival rates and the embryo recovery rates after 8 weeks of culture on proliferation medium. Explant survival rate was assessed as the frequency of explants showing signs of growth, including callus development. Embryo recovery rate was calculated as the percentage of explants showing resumption of somatic embryogenesis with evident cotyledonary embryos.

In plant regeneration experiments, the following parameters were estimated after culture of embryos on germination medium for 8 weeks: the percentage of embryos that simultaneously developed roots and shoots (plant conversion), as well as the root (mm) and shoot (mm) lengths and the number of leaves per recovered plant.

Data were analyzed by one or two-way analysis of variance (ANOVA I or II) and, when necessary, were transformed with the arcsine function. Untransformed data are presented in tables and figures. The post hoc analysis was evaluated by Dunnett’s method or by Least Significant Difference (LSD). A significance level of 0.05 was applied. All statistical analyses were performed using SPSS 26 software (SPSS Inc., Chicago, IL, USA).

## 5. Conclusions

In conclusion, the vitrification-based method presented here is simple, rapid and enables cryopreservation and regeneration of holm oak embryogenic cultures. Optimizing the PVS2 treatment time and the developmental stage of somatic embryos can significantly improve the embryo response to cryopreservation. Although the study findings demonstrate the feasibility of the long-term cryopreservation of holm oak embryogenic cultures, further research is needed to optimize the cryopreservation procedure, as genotype plays a key role in the success of the procedure.

## Figures and Tables

**Figure 1 plants-11-01266-f001:**
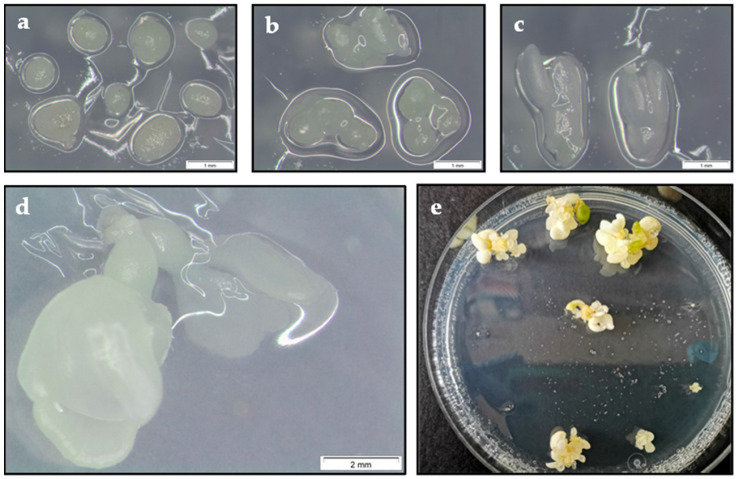
Cryopreservation of embryogenic cultures of holm oak by the vitrification-based method. a–c Show individual nodular embryogenic structures (**a**), groups of globular-heart stage embryos (**b**) and early cotyledonary stage embryos (**c**) isolated from line Q8. (**d**) Somatic embryos developed from a cryopreserved nodular embryogenic structure after 3 weeks of culture on proliferation medium. (**e**) Somatic embryo recovery from nodular embryogenic structures of line Q8 cryopreserved in liquid nitrogen for 4 years and cultured on proliferation medium for 8 weeks. Diameter of Petri dish, 90 mm.

**Figure 2 plants-11-01266-f002:**
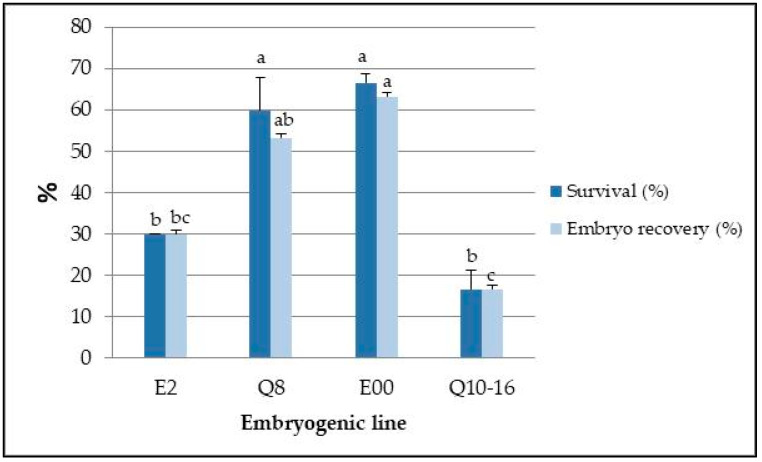
Survival and embryo recovery rates of four holm oak embryogenic lines after four years in LN. In each embryogenic line, columns represent means ± standard error of three replicates. Different letters on the bars indicate significant differences (at *p* = 0.05).

**Figure 3 plants-11-01266-f003:**
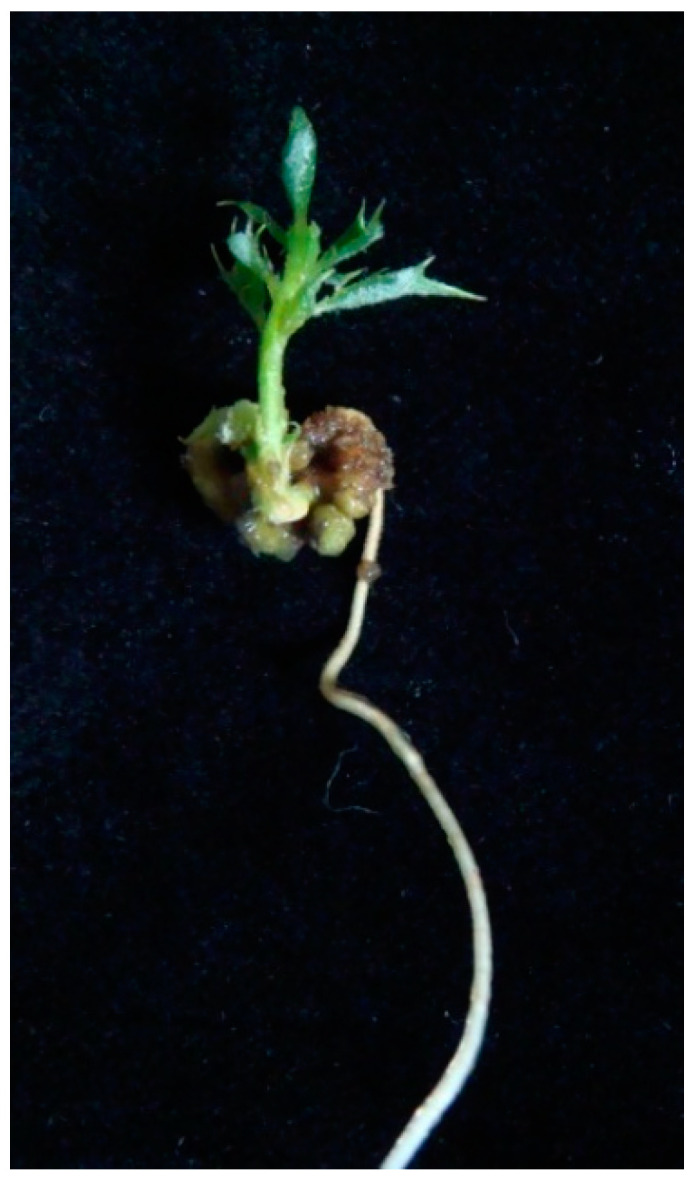
Somatic plant regenerated from embryogenic cultures of line Q8 after 15 min in PVS2, cryopreserved for one year and cultured for 8 weeks on germination medium.

**Table 1 plants-11-01266-t001:** Embryo survival (%) and recovery (%) in three different types of explant isolated from two holm oak embryogenic lines after preculture of embryos on medium containing sucrose and incubation in PVS2 solution for 15 min with (+LN) or without (−LN) and subsequent cryostorage in LN for one month.

Developmental Stage of Somatic Embryos	Survival (%)			Embryo Recovery (%)		
Line Q8	−LN	+LN	*Mean*	−LN	+LN	*Mean*
NES	100 ± 0.0 a	88.3 ± 2.8 a	94.1 a	96.7 ± 2.8 a	56.7 ± 2.8 b	*76.7 a*
Globular-heart	100 ± 0.0 a	3.3 ± 2.8 b	51.7 b	66.7 ± 2.8 b	3.3 ± 2.8 c	*35.0 b*
Cotyledonary	100 ± 0.0 a	0.0 ± 0.0 b	50.0 b	3.3 ± 2.8 c	0.0 ± 0.0 c	*1.65 c*
*Mean*	*100 a*	*30.5 b*		*55.6 a*	*30.0 b*	
**ANOVA II**						
Cryostorage in LN (A)	*p* ≤ 0.001			*p* ≤ 0.001		
Explant (B)	*p* ≤ 0.001			*p* ≤ 0.001		
A × B	*p* ≤ 0.001			*p* ≤ 0.01		
**Line E2**	−LN	+LN	*Mean*	−LN	+LN	*Mean*
NES	100 ± 0.0 a	80.0 ± 0.0 b	90.0 a	93.3 ± 2.8 a	46.7 ± 2.8 c	*70.0 a*
Globular-heart	100 ± 0.0 a	0.0 ± 0.0 c	50.0 b	66.7 ± 2.8 b	0.0 ± 0.0 d	*33.4 b*
Cotyledonary	100 ± 0.0 a	0.0 ± 0.0 c	50.0 b	3.3 ± 2.8 d	0.0 ± 0.0 d	*1.65 c*
*Mean*	*100 a*	*26.7 b*		*54.4 a*	*15.6 b*	
**ANOVA II**						
Cryostorage in LN (A)	*p* ≤ 0.001			*p* ≤ 0.001		
Explant (B)	*p* ≤ 0.001			*p* ≤ 0.001		
A × B	*p* ≤ 0.001			*p* ≤ 0.01		

Each value represents the mean ± standard error of three replicates. LN: liquid nitrogen; ns: not significant; NES: nodular embryogenic structures. Values indicated by different letters are significantly different (at *p* = 0.05).

**Table 2 plants-11-01266-t002:** (**a**) Survival (%) of nodular embryogenic structures of holm oak embryogenic lines (Q8 and E2) after preculture on medium containing sucrose and incubation in PVS2 solution for 15 or 30 min, with subsequent immersion in LN. (**b**) Recovery (%) of nodular embryogenic structures of holm oak embryogenic lines (Q8 and E2) after preculture on medium containing sucrose and incubation in PVS2 solution for 15 or 30 min, with subsequent immersion in LN.

**(a)**				
**Embryogenic Line/Time in PVS2 (min)**	**Control** **PVS2−LN**	**Storage Time in LN** **(Months)**		
**Q8**		1	6	12
PVS2 15	96.7 ± 3.2	90.0 ± 5.6	56.7 ± 3.2	63.3 ± 3.2
PVS2 30	90.0 ± 5.6	80.0 ± 0.0	60.0 ± 5.6	66.7 ± 3.2
**E2**		1	6	12
PVS2 15	90.0 ± 5.6	66.7 ± 8.5	63.3 ± 11.6	66.7 ± 3.2
PVS2 30	60.0 ± 5.6	53.3 ± 8.5	43.3 ± 3.2	40.0 ± 5.6
**ANOVA II**				
Genotype (A)	*p* ≤ 0.05	*p* ≤ 0.01	ns	ns
PVS2 time (B)	*p* ≤ 0.05	ns	ns	ns
A × B	ns	ns	ns	*p* ≤ 0.05
**(b)**				
**Embryogenic Line/Time in PVS2 (min)**	**Control ** **PVS2−LN**	**Storage Time in LN** **(Months)**		
**Q8**		1	6	12
PVS2 15	93.3 ± 3.2	83.3 ± 8.5	53.3 ± 3.2	60.0 ± 0.0
PVS2 30	86.7 ± 6.4	50.0 ± 5.6	60.0 ± 5.6	56.7 ± 8.5
**E2**		1	6	12
PVS2 15	90.0 ± 5.6	56.7 ± 11.6	53.3 ± 6.4	60.0 ± 5.6
PVS2 30	60.0 ± 5.6	40.0 ± 0.0	40.0 ± 0.0	40.0 ± 5.6
**ANOVA II**				
Genotype (A)	ns	ns	ns	ns
PVS2 time (B)	ns	*p* ≤ 0.05	ns	ns
A × B	ns	ns	ns	ns

Each value represents the mean ± standard error of three replicates. Controls were precultured on medium containing sucrose and vitrification treatment was applied, but the explants were not cryostoraged in liquid nitrogen (LN). ns: not significant.

**Table 3 plants-11-01266-t003:** Survival (%) and embryo recovery (%) of four holm oak embryogenic lines cryopreserved after preculture on medium containing sucrose and incubation in PVS2 solution for 15 min with (+LN) or without (−LN) subsequent immersion in LN for one month.

Embryogenic Line	Survival (%)		Embryo Recovery (%)	
	−LN	+LN	−LN	+LN
Q8	86.7 ± 3.2 a	80.0 ± 0.0 a	80.0 ± 6.4	60.0 ± 5.6
E2	86.7 ± 6.4 a	56.7 ± 3.2 b	80.0 ± 0.0	43.3 ± 8.5
E00	90.0 ± 0.0 a	40.0 ± 5.6 c	80.0 ± 0.0	36.7 ± 3.2
Q10–16	86.7 ± 6.4 a	46.7 ± 8.5 bc	80.0 ± 9.6	43.3 ± 6.4
*Mean*	87.5 ± 0.7 a	55.8 ± 7.6 b	80.0 ± 0.0 a	45.8 ± 4.3 b
**ANOVA II**				
Genotype (A)	ns		ns	
Cryostorage in LN (B)	*p* ≤ 0.001		*p* ≤ 0.001	
A × B	*p* ≤ 0.05		ns	

Each value represents the mean ± standard error of three replicates. LN: liquid nitrogen; ns: not significant. Values followed by different letters are significantly different (at *p* = 0.05).

**Table 4 plants-11-01266-t004:** Plant regeneration in two different holm oak embryogenic lines after treatment with PVS2 for 15 min, cryostorage for one year in LN and culture on germination medium.

Embryogenic Lines	Plant Regeneration (%)	Root Length (mm)	Shoot Length (mm)	Leaf Number
Q8 + LN	54.2 *±* 3.0	52.3 *±* 4.5	11.2 *±* 0.9	7.0 *±* 0.4
E2 + LN	45.8 ± 7.4	60.4 *±* 3.5	16.4 *±* 1.8	9.0 *±* 0.4

Each value represents the mean ± standard error of four replicates, each including 6 explants.

**Table 5 plants-11-01266-t005:** Nuclear DNA content (nDNA) of cryopreserved (+LN) and non-cryopreserved (−LN) embryos and plantlets regenerated from the corresponding embryogenic lines (E2 and Q8) of *Quercus ilex* (mean ± standard error).

Embryogenic Line	Tissue	Treatment	Ploidy	DNA Index	nDNA Content (pg/2C)	CV (%)	n	Statistics
**E2**								
	Embryos	−LN	2C	0.244 ± 0.002	2.22 ± 0.02	5.27	4	ns
		+LN	2C	0.243 ± 0.008	2.12 ± 0.07	5.68	4	ns
	Plantlets	−LN	2C	0.237 ± 0.006	2.15 ± 0.06	5.19	4	ns
		+LN	2C	0.247 ± 0.005	2.24 ± 0.04	5.41	2	ns
**Q8**								ns
	Embryos	−LN	2C	0.216 ± 0.004	1.97 ± 0.04	5.27	4	ns
		+LN	2C	0.219 ± 0.003	1.99 ± 0.03	5.68	4	ns
	Plantlets	−LN	2C	0.226 ± 0.004	2.06 ± 0.04	5.19	3	ns
		+LN	2C	0.233 ± 0. 007	2.12 ± 0.07	5.41	3	ns

LN: liquid nitrogen; n: sample number; ns: not significant.

## Data Availability

Not applicable.

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
