# Peer review of "Cryopreservation of Holm Oak Embryogenic Cultures for Long-Term Conservation and Assessment of Polyploid Stability"

_plants, 2022, doi:10.3390/plants11091266_

Round 1

Reviewer 1 Report

Dear Authors,

According to me, the article is well written, concise and pleasant to read. The analysis looks coherent and well organised as well, with a deep commentary of results. Conclusions are supported by the analyses.

The article is good and almost ready for publication

  1. All tables - I think it would be better to write the text in lower case.
  2. Table 3 - EMBRYO RECOVERY column - missing statistics?

Author Response

See file.

Reviewer 2 Report

The authors provide a vitrification protocol for long-term cryopreservation of holm oak. The work examines somatic embryos (SE) as the storage entity and refines the best choice to a particular developmental stage of the SE. A 2-factor design examining SE stage and liquid nitrogen storage with survival and embryo recovery as observations. Nodular embryo stages were further examined in a factorial design involving 2 genotypes and 2 cryogenic incubation periods. Comparisons to controls are noted to be done using Dunnett's test. Four year, long-term data are also shown for different genotypes using the factor combinations giving highest survival and recovery metrics.

The work is well described and the use of replicated factorial designs is appropriately done. Additional information on plant regeneration and genetic stability confirm the value of the approach. Factorial designs are often a prelude to response surface designs that allow for truer optima that are not located on the boundaries of the factor values. I would avoid using the 'optimal' designation and perhaps choose 'best' or 'highest'.

There are no significant changes required. The manuscript currently has an annoying page break for Table 2A, lines 146 and 147, separating the Table title from the table entries.

Author Response

See file.
